# Preventive Measures for SARS-CoV-2 in the Workplace and Vaccine Acceptance: Assessment of Knowledge, Attitudes and Behaviors of Workers in Southern Italy

**DOI:** 10.3390/vaccines10111872

**Published:** 2022-11-05

**Authors:** Concetta Paola Pelullo, Pamela Tortoriello, Livio Torsiello, Chiara Lombardi, Francesco Napolitano, Gabriella Di Giuseppe

**Affiliations:** 1Department of Movement Sciences and Wellbeing, University of Naples “Parthenope”, 80133 Naples, Italy; 2Department of Experimental Medicine, University of Campania “Luigi Vanvitelli”, 80138 Naples, Italy

**Keywords:** SARS-CoV-2, COVID-19 vaccine, preventive measures, workplace, survey

## Abstract

(1) Background: this study investigated the preventive measures implemented in the workplace and evaluated knowledge, attitudes and adherence behaviors regarding SARS-CoV-2 routes of transmission and preventive measures in a group of workers. (2) Methods: this cross-sectional study was conducted from May to July 2021 among 501 workers in the Campania region, in Southern Italy. (3) Results: 80.5% of respondents declared that their company had implemented the main COVID-19 preventive measures, and 54.7% of respondents knew SARS-CoV-2 routes of transmission along with the main preventive measures. Moreover, 34.2% were highly concerned about contracting SARS-CoV-2 in the workplace and transmitting it to family. Adherence to all preventive measures in the workplace involved 42.5% of respondents. The results of the multivariate logistic regression model revealed that significant determinants of adherence to all preventive measures in the workplace were being female, working not as a manager or office employee, cohabiting with someone that received a diagnosis of SARS-CoV-2 infection, knowing SARS-CoV-2 routes of transmission and the related main preventive measures, being highly concerned of contracting SARS-CoV-2 in the workplace and transmitting it to family and believing that COVID-19 vaccine offers high protection against disease. At the time of the survey, 47.5% of respondents had already received COVID-19 vaccine. Among unvaccinated respondents, 11.8% expressed unwillingness to get vaccinated for COVID-19. (4) Conclusions: These findings highlighted a good awareness about COVID-19 prevention and underlined a good propensity to get vaccinated among workers. Therefore, there is the need that preventive measures should be prioritized in the working context.

## 1. Introduction

The COVID-19 pandemic has had a relevant impact on working activities all around the globe, stimulating the development and implementation of workplace safety guidelines in several countries, including in the USA by the Occupational Safety and Health Administration (OSHA) [1] and in Europe by the European Agency of Safety and Health at Work [2]. Moreover, great attention has been paid to understand the dynamics of the spread of the virus in work contexts [3], and the economic and health consequences of infection in prevention and control activities prompted in work settings [4,5,6,7].

In Italy, on 6 April 2021, a shared protocol was signed by the Italian government to update previous measures (April 2020) in order to successfully contain the spread of the severe acute respiratory syndrome coronavirus 2 (SARS-CoV-2) in workplaces [8]. The document contained guidelines aimed at enabling the adoption of coronavirus disease 2019 (COVID-19) prevention and control protocols in workplace settings.

Moreover, the availability of COVID-19 vaccines prompted the Ministry of Health and the Ministry of Labor, in agreement with the Conference of Regions, the Extraordinary Commissioner for the fight against the COVID-19 emergency and the technical/scientific contributions of the National Institute for Insurance against Accidents at Work (INAIL), to prepare a specific document containing: “Interim indications for anti-SARS-CoV-2/COVID-19 vaccination in workplaces” to be applied throughout the country for the establishment and the management of extraordinary and temporary vaccination centers in the workplaces [9]. This initiative allowed companies to vaccinate their staff in the workplace.

Furthermore, according to the National Prevention Plan 2020–2025, people can be more easily reached in the workplace to be involved in health promotion [10], so the implementation of workplace health promotion (WHP) can produce benefits in terms of workers’ health; therefore, particularly in the context of the COVID-19 pandemic, prevention and promotion of health in the workplace has assumed even more importance. Moreover, based on the document of the Minister of Labor containing: “Interim indications for anti-SARS-CoV-2/COVID-19 vaccination in the workplace”, the vaccination of workers contributes to the continuation of commercial and production activities by increasing the level of safety in workplaces in addition to the fact that sufficient vaccination coverage can only be obtained by knowing the determinants of vaccine hesitancy. 

Several studies have evaluated the application of anti-contagion directives in workplaces and the workers’ compliance with these regulations [11,12,13] as well as the acceptability of COVID-19 vaccine among various professional categories [14,15,16], whereas, to the best of our knowledge, there is the absence in the Italian literature of a study that evaluates the anti-contagion directives in the workplace and compliance with these regulations by workers. Therefore, the objectives of the current study were to investigate the preventive measures implemented in the workplace by the recruited companies; to evaluate knowledge, attitudes and adherence behaviors regarding SARS-CoV-2 routes of transmission and preventive measures; and to investigate the willingness to vaccinate with COVID-19 vaccine and the key predictors of this intention among a sample of workers in Southern Italy.

## 2. Materials and Methods

### 2.1. Study Design and Sample Size Planning

This cross-sectional study was conducted from May to July 2021 among workers in the geographic area of the Campania region in Southern Italy.

From a list of companies located in the provinces of Naples and Caserta, those with a number of employees greater than 10 and involved in the agri-food, engineering or commercial sectors were considered eligible. Thereafter, 10 companies were randomly selected and all workers who were not working from home at the time of the survey were asked to complete a self-administered survey.

The sample size was calculated with the assumption of a response rate of 75%, a prevalence of adherence to preventive measures of 65.5%, a confidence interval of 95% and an error of 5%. The minimum sample size required was estimated to be 347 subjects.

### 2.2. Participants and Data Collection

Anonymous questionnaires were delivered to workers after receiving the employer’s approval during a preliminary meeting. The objectives and the methodology of the study were explained to the employer and to all participants, who gave informed consent before the participation. No incentives were offered to participants.

Compiled questionnaires were collected by the staff of the Public Health Section of the Department of Experimental Medicine of the University of Campania “Luigi Vanvitelli”.

### 2.3. Survey Instrument

The questionnaire was constructed ad hoc through an extensive literature review, based on items of previous investigations regarding vaccinations on other populations [17,18,19,20,21] and workers [12,14,16].

The survey consisted of four sections: first, there were open and multiple choice questions about age, gender, nationality, marital status, number of sons/daughters, number of cohabitants, education level, employment type, weekly working hours, night shifts, working from home during pandemic, years of working activity, SARS-CoV-2 infection history of the respondent and cohabitants, preventive measures implemented by the company, screening/diagnostic SARS-CoV-2 tests performed in the workplace and comorbidities. The self-reported health status was measured on a 10-point Likert scale, with values ranging from 1 (low) to 10 (high). The second part contained questions with multiple-choice response format regarding knowledge and attitudes of workers about SARS-CoV-2 routes of transmission, symptoms and preventive measures; they were measured on a 3-point Likert-type scale with responses ranging from “agree” to “disagree” or on a 10-point Likert-type scale with values ranging from 1 to 10. The third part’s questions, regarding behaviors about preventive measures, flu and COVID-19 vaccination, were close-ended with “yes” or “no” or multiple choice or on 5-point Likert-type scale with “never”, “rarely”, “sometimes”, “often” and “always” response format. Finally, the fourth part contained four questions concerning sources of information and need for information about COVID-19 preventive measures in the workplace and COVID-19 vaccination, with a multiple choice response format and open questions.

The study was preceded by a pilot test performed among 50 workers in order to evaluate the readability and clarity of the questionnaire. The pilot study data were added to the general database. 

### 2.4. Statistical Analysis

The data were analyzed using Stata software version 15 [22]. To describe the sample characteristics, descriptive statistics were performed, using means, frequencies and standard deviations. The bivariate analysis was carried out by means of the chi-squared test and Student’s *t*-test for categorical and continuous variables, respectively.

The multivariate models were conducted in order to identify all factors influencing the following outcomes of interest: knowledge of SARS-CoV-2 routes of transmission and preventive measures (no = 0; yes = 1) (Model 1); high concern of contracting and transmitting SARS-CoV-2 to family (no = 0; yes = 1) (Model 2); adherence to all preventive measures in the workplace (no = 0; yes = 1) (Model 3); unwillingness to get vaccinated for COVID-19 because the vaccination was considered not useful and/or safe and/or effective (no = 0; yes = 1) (Model 4).

In Model 1, the outcome was dichotomized as follows: workers who knew all SARS-CoV-2 routes of transmission (through saliva, touching mouth, nose or eyes, personal direct contacts) and those who knew all main preventive measures (use of face masks, frequent hand washing, avoiding crowded places, social distancing) versus all others. In Model 2, the outcome investigating concern of contracting and transmitting SARS-CoV-2 to family, originally structured in 10-point Likert scale format, was dichotomized as follows: 1–7 (low) = 0 and 8–10 (high) = 1. In Model 3, the outcome was dichotomized as follows: workers who always wore masks, those who always practiced hand antisepsis and those who always applied physical distancing versus all others. The fourth outcome investigating unwillingness to get vaccinated for COVID-19 was constructed by separating, among those not yet vaccinated, those who considered COVID-19 vaccine not useful and/or safe and/or effective versus all others.

The independent variables included in logistic regression models are reported in an additional file (Appendix A: Variables included in the logistic regression models with related categories).

## 3. Results

### 3.1. Characteristics of Selected Companies

The 10 companies selected to take part in this survey, located in the provinces of Naples and Caserta, were small (4 companies from 10 to 50 employees), medium (5 companies from 50 to 250 employees) and large ones (1 company with more than 250 employees) and belonged to the agri-food (4 companies), engineering (3 companies) or commercial sectors (3 companies).

### 3.2. Participants’ Characteristics

Of the 560 questionnaires distributed, 501 were returned for an overall response rate of 89.5%. The characteristics of the respondents are summarized in Table 1. Two-thirds of the sample were males (66.3%); age ranged from 20 to 64 with an average of 40.1; 29.3% of the respondents had a university degree; and half were employed as factory workers (46.8%). Average self-rated health status was 8.4 out of a score of 1 to 10, 13% of the respondents suffered from at least one chronic disease and 13.4% had been infected with SARS-CoV-2.

### 3.3. Main Preventive Measures Implemented by Companies

The main preventive measures implemented by companies are summarized in Table 2; 80.5% of respondents declared that their company had implemented the main COVID-19 preventive measures in the workplace, such as body temperature measurement at the entrance (98.2%), availability of hand sanitizer dispensers (99.6%), free mask distribution (80.6%), disinfection of indoor areas (85.6%), adequate distance among workstations (94.4%) and indoor ventilation (86.8%). Moreover, 42.7% of respondents declared that they had never undergone screening/diagnostic SARS-CoV-2 tests in the workplace.

### 3.4. Participants’ Knowledge

The respondents’ knowledge about COVID-19 is described in Table 3. Overall, 89.7% knew that saliva (respiratory droplets) was one of the routes of transmission, 90.9% reported contaminated hands, 65.6% knew that transmission of the virus could occur by direct contact with infected people, 59.9% knew all three routes of transmission and 37.3% of the respondents knew the main three symptoms (fever, cough, asthenia) of COVID-19.

Knowledge of all main SARS-CoV-2 preventive measures was reported by 80.1% of the participants, specifically use of masks by 98.6%, hand hygiene by 96.5%, avoiding crowded places by 95.1% and physical distancing by 88.4%.

Moreover, 54.7% of respondents knew both the SARS-CoV-2 routes of transmission and preventive measures. In the logistic regression analysis, factors significantly influencing a higher knowledge about SARS-CoV-2 routes of transmission and main anti-COVID-19 preventive measures were older age (OR = 1.03; 95% CI = 1.001–1.06), female gender (OR = 1.82; 95% CI = 1.13–2.93), university degree (OR = 1.74; 95% CI = 1.05–2.88), low self-rated health status (OR = 0.57; 95% CI = 0.34–0.94) and not working from home during the pandemic (OR = 0.47; 95% CI = 0.27–0.82) (Table 4, Model 1).

### 3.5. Participants’ Attitudes

The respondents’ attitudes about main preventive measures and COVID-19 vaccination are described in Table 3. Use of masks (95.4%), physical distancing (90.8%) and hand hygiene (96%) were perceived to reduce the risk of SARS-CoV-2 transmission in the workplace. Moreover, 73.3% believed that COVID-19 vaccine protects their family from contagion and 74.1% felt that a vaccine offers high protection from COVID-19. Finally, 34.2% was highly concerned about contracting SARS-CoV-2 in the workplace and transmitting it to family and 42.8% considered COVID-19 vaccine highly useful and safe. 

Fear of contracting SARS-CoV-2 in the workplace and transmitting it to family was significantly associated with working not as manager or office employee (OR = 0.35; 95% CI = 0.21–0.61), with knowledge of SARS-CoV-2 routes of transmission and main preventive measures (OR = 1.98; 95% CI = 1.19–3.29) and with having had a cohabitant infected with SARS-CoV-2 (OR = 2.23; 95% CI = 1.16–4.28) (Table 4, Model 2).

### 3.6. Participants’ Behavior

Physical distancing in the workplace was reported to always be respected by 61.5%, 80.6% always used masks, 58.5% used hand sanitizer dispensers, 57.8% used body temperature measurement devices, 74.8% always respected entry and exit signs, only 5.7% always used public transportation to go to the workplace and 35.8% always avoided crowded places outside of work.

Only 42.5% of respondents respected all preventive measures in the workplace. In the multivariate logistic regression analysis, factors associated with adherence to all preventive measures in the workplace were being female (OR = 2.94; 95% CI = 1.60–5.39), working not as manager or office employee (OR = 0.32; 95% CI = 0.17–0.59), cohabiting with someone that received a diagnosis of SARS-CoV-2 infection (OR = 2.90; 95% CI = 1.22–6.84), knowing SARS-CoV-2 routes of transmission and related main preventive measures (OR = 3.50; 95% CI = 1.97–6.19), being highly concerned about contracting SARS-CoV-2 in the workplace and transmitting it to family (OR = 2.07; 95% CI = 1.14–3.76) and believing that COVID-19 vaccine offers high protection against disease (OR = 2.41; 95% CI = 1.09–5.30) (Table 5, Model 3).

About one-fifth of the respondents (17.9%) had received a flu vaccination in the 2020–2021 influenza season and 47.5% were against COVID-19 at the moment of survey. Reasons for COVID-19 vaccination uptake were: belonging to the categories already convened at the time of the survey (49.1%), perceiving to be at risk for COVID-19 (26.1%), believing that COVID-19 vaccine is effective (12.3%), being not worried about the side effects of the COVID-19 vaccine (8.9%), believing that COVID-19 vaccine is safe (5.7%) and perceiving a sense of responsibility towards the community (2.7%). Reported reasons for not being vaccinated for COVID-19 vaccination were: not belonging to the categories already convened at the time of the survey (78.9%), waiting to be convened (9.3%), being worried about the side effects of the COVID-19 vaccine (7.3%), not believing that COVID-19 vaccine is safe (3.7%), not perceiving to be at risk for COVID-19 (3.2%), not believing that COVID-19 vaccine is effective (2.4%) and not being entitled by age at the time of the survey (2.4%).

At the time of the survey, among workers who were not yet vaccinated, 11.8% were not willing to get vaccinated for COVID-19 because they did not consider the vaccine useful and/or safe and/or effective. Multiple logistic regression analyses showed that being older (OR = 1.10; 95% CI = 1.01–1.22), males (OR = 0.05; 95% CI = 0.004–0.61), those with worse self-rated health status (OR = 0.13; 95% CI = 0.02–0.80), those with fewer years of working activity (OR = 0.84; 95% CI = 0.74–0.95), not believing that physical distancing in the workplace reduces the risk of SARS-CoV-2 transmission (OR = 0.02; 95% CI = 0.002–0.19), not believing that the vaccine offers high protection against COVID-19 (OR = 0.08; 95% CI = 0.01–0.57) and needing further information on COVID-19 prevention or the vaccine (OR = 11.55; 95% CI = 1.48–89.84) and not being willing to get vaccinated for COVID-19 because the vaccination was considered not useful and/or safe and/or effective (Table 5, Model 4).

Among those who had not yet been vaccinated for COVID-19 and that expressed their refusal to be vaccinated in the future, they reported reasons that they did not believe the COVID-19 vaccine was useful, safe and/or effective.

### 3.7. Participants’ Main Sources of Information

Questions regarding sources of information indicated that 64% of workers acquired knowledge on preventive measures from media, 58.3% from companies, 19.2% from physicians, 5.9% from trade associations and 0.4% from health and safety managers. Information about the COVID-19 vaccine was acquired by 68.8% of participants from media, 43.2% from physicians and 1.7% from companies. Respectively, 6.6% and 10.8% of the respondents would have liked to receive more information about COVID-19 prevention and vaccination in the workplace.

## 4. Discussion

To our knowledge, this is the first Italian study assessing the adherence to anti-COVID-19 vaccination among workers after the release of vaccines for the general population. Furthermore, the study explored the application of main preventive measures for the containment of contagion in the workplace.

At the time of the survey, 47.5% of workers had already received at least the first dose of the anti-COVID-19 vaccine, and 91.9% of respondents declared they would be vaccinated as soon as the vaccine was available for their category. A similar study conducted in China by Zhang et al. in 2021 showed that the prevalence of vaccination acceptability among workers in some Chinese companies was 66.6% and rose to 80.6% if vaccines were made available free of charge [14]. Another study conducted in China by Wang et al. estimated the prevalence of vaccine acceptability during both the first wave (February 2020) and the third wave of the local epidemic (August–September 2020). Working people were found to be more willing to accept the vaccine (44.2% vs. 34.8%) and less likely to be hesitant (38.6% vs. 43.7%) in the first wave than in third [16]. A cross-sectional survey conducted in China from March to April 2021 investigated the acceptance of the COVID-19 vaccine among a sample of industrial workers, with 66.0% of participants expressed willingness to receive COVID-19 vaccine, while 16.6% expressed resistance and 17.4% insecurity [23]. Another study carried out by Wang et al. through an anonymous online questionnaire among non-healthcare and healthcare workers in January 2021 showed that 56.19% of non-healthcare workers received the COVID-19 vaccine, 37.57% were hesitant and 6.24% were resistant. Vaccine-resistant individuals were also more likely to be female and older than 65 years. Similarly, in our research, the workers who expressed unwillingness to get vaccinated for COVID-19 because the vaccination was considered not useful and/or safe and/or effective were older the ones [24]. Liu et al. conducted a cross-sectional COVID-19-related knowledge, attitudes, and practices survey in China during December 2020 among cold-chain workers: 76% indicated that they were willing to be vaccinated. In this study knowledge about SARS-CoV-2 (comprehending the most effective prevention, understanding the transmission routes and recognizing the priority vaccination groups) were positively associated with willingness to be vaccinated against COVID-19 [25]; similarly, our survey found that workers who had a lower knowledge of all SARS-CoV-2 routes of transmission and main preventive measures were less likely to be vaccinated. During the early stages of the pandemic, when the vaccine was not yet available, a study conducted by La Vecchia et al. in September 2020 estimated the prevalence of vaccine acceptability to be 51.6% among professionals and teachers and to be 44.8% among manual workers and farmers [26]. The differences found concerning willingness could depend on various factors: (1) free access to vaccination in the Italian territory and (2) an increase in confidence in vaccination. At the time of the survey, the vaccination campaign had started several months before and this may have contributed to the increased confidence in vaccination.

The second objective of this study was to assess workers’ adherence to the anti-COVID-19 preventive measures and the large majority (80.1%) of the respondents recalled the main ones. During the first pandemic phase (March 2020), a cross-sectional study was conducted in China by Pan et al. [12] with the aim of showing compliance with the four main anti-COVID-19 preventive measures in the workplace. The data from this study showed that 96.8% of the respondents always wore a face mask in the workplace, 70.9% sanitized their hands and 65.8% avoided crowded places [12]. Another cross-sectional online survey conducted in March 2020 in China by Pan et al. showed that 95.7% of participants reported consistent face mask wearing in any public places, and 70.9% sanitized hands every time after returning from public spaces or touching installations [27]. Our study found a different compliance among Italian workers: 80.6% always used face masks, 58.5% always sanitized their hands and 35.8% avoided crowded places outside the workplace. In the study conducted by Zhang et al. in September 2020 [14], 81.6% of the workers reported that they always wore a mask outside working hours, 43.3% of the respondents reported practicing physical distancing with non-cohabitants and 59.3% reported that they always sanitized their hands after returning from public spaces or touching public installations [14].

It may be plausible that the differences found with respect to the Chinese surveys lie in the different timeframe in which the studies were conducted. The present study was carried out at a more advanced stage of the pandemic, when the vaccines had already been released for the risk categories and had recently been opened also to the general population. In particular, the findings of Zhang et al. [14] highlighted vaccine acceptability, projected in the future in terms of hypothetical uptake, while our study analyzed the vaccination status, evaluating the relative attitudes of the vaccinated and unvaccinated respondents. In our study, having received seasonal influenza vaccination was positively correlated with having received anti-COVID-19 vaccine, just as expressed by Zhang et al. in regard to the willingness of receiving anti-COVID-19 vaccine and influenza vaccination status. Different findings emerged with having had a family member with COVID-19 or having sons/daughters, where, according to Zhang, these factors correlated with a higher intention to receive COVID-19 vaccination, whereas these factors were not associated in our study in a statistically significant way. In accordance with the Zhang study, the high behavioral intention to receive a COVID-19 vaccine of the factory workers was confirmed in our study as a good vaccine uptake [14].

In our study, workers were asked which main preventive measures were applied by companies. According to the results of this study, the main preventive measures enforced in the workplace were body temperature measurement, free mask distribution, hand sanitizer dispensers, disinfection of indoor areas and distance among workstations, whereas less than half (42.7%) of respondents had not received a screening/diagnostic SARS-CoV-2 test in the workplace. Wearing masks and temperature measurement in the workplace were among the most prevalent workplace measures also reported in previous investigations [12,28,29]. The companies enrolled have adhered to the directives contained in the document of the Minister of Labor regarding the main preventive measures to be adopted in order to increase the level of safety in workplaces. In addition, 57.3% of workers had performed at least one swab on the initiative of the employers since the start of the pandemic, despite the guidelines not mentioning the need to perform screening/diagnostic SARS-CoV-2 tests in the workplace.

One of the strengths of this study is that, through the questionnaire, it was possible to explore not only knowledge and attitudes about COVID-19 but also behaviors regarding the application of preventive measures by workers and companies in the light of the two protocols issued by the Ministry of Labor and Social Policies and the Minister of Health in April 2021. Another strength of the study was that all questionnaires were responded to in a safe environment, where direct pressure from employers to give untruthful answers was minimal.

This survey has some intrinsic limitations of cross-sectional studies, such as the inability to determine temporal sequentiality and to assess causal relationships. Moreover, social desirability bias is possible, caused by the tendency of respondents to give socially desirable answers instead of choosing answers that would reflect their true intentions and behaviors. To minimize this, the questions were formulated in a way that did not influence in any way the respondents, especially in regard to reasons behind personal vaccination status. Recall bias and telescoping bias in particular could have influenced the results in unpredictable manners. Additionally, it was not properly addressed for questionnaire limitations, involving the relation between employer and worker concerning mutual effort to limit the spreading of SARS-CoV-2 in the workplace. Moreover, this study was carried out in one geographical area of Southern Italy and, therefore, this can affect the generalizability of the results to the whole of Italy. However, we are confident that the characteristics of the participants of the survey are like those of other geographical areas of Italy.

## 5. Conclusions

In conclusion, this study highlighted both a good awareness about COVID-19 prevention and general knowledge. Regarding vaccination for SARS-CoV-2, most workers showed attentiveness and knowledge, underlining a good propensity to get vaccinated. In this context, it is strategically important to regain those workers who expressed unwillingness to receive COVID-19 vaccine and to improve their adherence to preventive measures. For this purpose, designing dedicated vaccination hubs in the workplace could be a major strategy for a greater adherence and also a good source of prevention information for workers.

In light of this, considering the growing necessity of the general population for health and safety during a pandemic and to appropriately prepare for possible future emergencies, prioritizing implementation of preventive measures in the workplace, including a solid vaccination policy, should be considered. 

## Figures and Tables

**Table 1 vaccines-10-01872-t001:** Socio-demographic and anamnestic characteristics of the study population (N = 501).

CHARACTERISTICS		
*Socio-Demographic*	N	%
Gender ª		
Male	326	66.3
Female	166	33.7
Age, in years	40.07 ± 9.7 (20–64) *	
Nationality		
Italians	482	96.2
Others	19	3.8
Marital status ª		
Married	302	60.6
Other	196	39.4
Sons/daughters ª		
None	188	37.6
≥1	312	62.4
Number of cohabitants ª		
None	30	6.2
≥1	458	93.8
Education level		
Other	354	70.7
University degree	147	29.3
Working activity		
Employment type ª		
Manager/office employee	193	39.3
Other	298	60.7
Weekly working hours ª	39.7 ± 6.6 (5–80) *	
Night shifts ª		
No	420	84.5
Yes	77	15.5
Number of night shifts in a month ª	6.3 ± 2.6 (1–10) *	
Working from home during pandemic ª		
No	356	71
Yes	145	29
Years of work ª	10.3 ± 8.9 (0–43) *	
≤10	289	58.7
>10	203	41.3
Having contractedSARS-CoV-2 infection ª		
No	432	86.6
Yes	67	13.4
Having had cohabitants infected with SARS-CoV-2 ª		
No	394	79.4
Yes	102	20.6
Suffering from at least one chronic disease ª		
No	433	87
Yes	65	13
Cardiovascular diseases	38	62
Metabolic diseases	10	16.4
Autoimmune diseases	7	11.5
Respiratory diseases	5	8.2
Other	4	6.6
Genitourinary diseases	3	4.9
Taking medications for underlying chronic clinical conditions ª^,^º		
No	21	30.9
Yes	47	69.1
Self-rated health status ª	8.4 ± 1.32 (2–10) *	
Low (1–7)	122	25.8
High (8–10)	351	74.2
Flu vaccination uptake in the 2020–21 influenza season ª		
No	400	82.1
Yes	87	17.9
COVID-19 vaccine uptake ª		
No	258	52.6
Yes	233	47.4

* Mean ± standard deviation (range); ª number for each item may not add up to total number of study population due to missing values; º only among those who have chronic diseases.

**Table 2 vaccines-10-01872-t002:** Preventive measures provided by companies (N = 501).

MAIN PREVENTIVE MEASURES		
	N	%
Body temperature measurement at the entrance ª		
Yes	491	98.2
No	9	1.8
Hand sanitizer dispensers		
Yes	499	99.6
No	2	0.4
Free masks distribution		
Yes	404	80.6
No	97	19.4
Disinfection of indoor areas ª		
Yes	427	85.6
No	72	14.4
Distance among workstations ª		
Yes	471	94.4
No	28	5.6
Indoor ventilation ª		
Yes	434	86.8
No	66	13.2
Screening/diagnostic SARS-CoV-2 tests ª		
No	210	42.7
Less than once a month	103	20.9
One or more times a month	80	16.3
Yes (without specifying when)	56	11.4
For necessity	43	8.7

ª Number for each item may not add up to total number of study population due to missing values.

**Table 3 vaccines-10-01872-t003:** Knowledge and attitudes about COVID-19: transmission, main symptoms and preventive measures (N = 501).

KNOWLEDGE ABOUT COVID-19	Total	
Routes of transmission ª	N	%
Saliva	443	89.7
Touching mouth, nose or eyes	449	90.9
Personal direct contacts	324	65.6
All three modes of transmission	296	59.9
Frequent symptoms ª		
Fever/Cough/Tiredness	179	37.3
Others	301	62.7
Main preventive measures ª		
Use of face masks	486	98.6
Frequent hand washing	475	96.5
Avoiding crowded places	429	95.1
Physical distancing	435	88.4
All four main preventive measures	394	80.1
Belief that use of masks in the workplace reduces risk of SARS-CoV-2 transmission ª		
Agree	476	95.4
Uncertain/Disagree	23	4.6
Belief that physical distancing in the workplace reduces risk of SARS-CoV-2 transmission ª		
Agree	453	90.8
Uncertain/Disagree	46	9.2
Belief that hand hygiene in the workplace reduces risk of SARS-CoV-2 transmission ª		
Agree	479	96
Uncertain/Disagree	20	4
Belief that the COVID-19 vaccine also protects family from contagion ª		
Agree	366	73.3
Uncertain/Disagree	133	26.7
Belief that the vaccine offers high protection against COVID-19 ª		
Agree	355	74.1
Uncertain/Disagree	144	25.9
Being highly concerned about contracting and transmitting SARS-CoV-2 to family ª		
Yes	145	34.2
No	279	65.8
Belief that the COVID-19 vaccine is highly useful and safe ª		
Yes	209	42.8
No	279	57.2

ª Number for each item may not add up to total number of study population due to missing values.

**Table 4 vaccines-10-01872-t004:** Logistic regression model for potential determinants of the outcome of interest (N = 501).

Variable	OR *	SE **	95% CI ◦	*p* Value
Model 1. Knowledge of SARS-CoV-2 Routes of Transmission and Preventive Measures (N = 267, n = 54.7%)				
Log likelihood = −272.26; χ2 = 30.59 (8 df); *p* = 0.0002				
*Working from home during pandemic*				
No	1 ª			
Yes	0.47	0.13	0.27–0.82	0.008
*Gender*				
Male	1 ª			
Female	1.82	0.44	1.13–2.93	0.013
*Education level*				
Other	1 ª			
University degree	1.74	0.44	1.05–2.88	0.029
*Self-rated health status*				
Low (1–7)	1 ª			
High (8–10)	0.57	0.14	0.34–0.94	0.029
*Age, in years (continuous)*	1.03	0.015	1.01–1.06	0.042
*Years of working activity* *(continuous)*	0.97	0.01	0.94–1.01	0.105
*Employment type*				
Other	1 ª			
Manager/Office employee	1.44	0.36	0.87–2.38	0.153
*Having received information about COVID-19 prevention in the workplace from companies*				
No	1 ª			
Yes	1.22	0.26	0.80–1.85	0.348
**Model 2. High Concern about Contracting and Transmitting SARS-CoV-2 to Family (N = 145, n = 34.2%)**	**OR ***	**SE ****	**95%CI ◦**	***p* Value**
Log likelihood = −197.33; χ2 = 38.80 (7 df); *p* < 0.0001				
*Employment type*				
Other	1 ª			
Manager/Office employee	0.35	0.09	0.21–0.61	<0.001
*Knowledge of all SARS-CoV-2 routes of transmission and main preventive measures*				
No	1 ª			
Yes	1.98	0.51	1.19–3.29	0.008
*Having had cohabitants infected withSARS-CoV-2*				
No	1 ª			
Yes	2.23	0.74	1.16–4.28	0.016
*Age in years (continuous)*	1.02	0.01	0.99–1.05	0.077
*Self-rated health status*				
Low (1–7)	1 ª			
High (8–10)	0.62	0.18	0.35–1.11	0.112
*Having received information about COVID-19 prevention in the workplace from companies*				
No	1 ª			
Yes	0.75	0.19	0.45–1.24	0.268
*Gender*				
Male	1 ª			
Female	1.35	0.39	0.76–2.39	0.291

ª Reference category; * Odds ratio; ** standard error; ◦ confidence interval.

**Table 5 vaccines-10-01872-t005:** Logistic regression model for potential determinants of the outcome of interest (N = 501).

**Model 3. Adherence to All Preventive Measures in the Workplace (N = 210, n = 42.5%)**	**OR ***	**SE ****	**95%CI ◦**	***p* Value**
Log likelihood = −160.12; χ2 = 125.78 (10 df); *p* < 0.0001				
*Gender*				
Male	1 ª			
Female	2.94	0.91	1.60–5.39	<0.001
*Employment type*				
Other	1 ª			
Manager/Office employee	0.32	0.101	0.17–0.59	<0.001
*Knowledge of all SARS-CoV-2 routes of transmission and main preventive measures*				
No	1 ª			
Yes	3.50	1.02	1.97–6.19	<0.001
*Belief that the COVID-19 vaccine is highly useful and safe*				
Low (1–7)	1 ª			
High (8–10)	3.08	0.93	1.69–5.59	<0.001
*Having had cohabitants infected withSARS-CoV-2*				
No	1 ª			
Yes	2.90	1.27	1.22–6.84	0.015
*Being highly concerned about contracting and transmitting SARS-CoV-2 to family*				
No	1 ª			
Yes	2.07	0.63	1.14–3.76	0.016
*Having received information about COVID-19 prevention in the workplace from companies*				
No	1 ª			
Yes	2.00	0.59	1.11–3.59	0.020
*Belief that the vaccine offers high protection against COVID-19*				
Uncertain/Disagree	1 ª			
Agree	2.41	0.96	1.09–5.30	0.028
*Belief that the COVID-19 vaccine also protects family from* *contagion*				
Uncertain/Disagree	1 ª			
Agree	1.82	0.70	0.86–3.88	0.116
*Need for further information on COVID-19 prevention or COVID-19 vaccine*				
No	1 ª			
Yes	0.62	0.26	0.27–1.42	0.260
**Model 4. Unwillingness to Get Vaccinated for COVID-19 Because the Vaccination was Considered Not Useful and/or Safe and/or Effective (N = 29, n = 11.8%)**	**OR ***	**SE ****	**95%CI ◦**	***p* Value**
Log likelihood = −30.72; χ2 = 53.86 (12 df); *p* < 0.0001				
*Belief that physical distancing in the workplace reduces risk of SARS-CoV-2 transmission*				
Uncertain/Disagree	1 ª			
Agree	0.02	0.02	0.002–0.19	0.001
*Years of working activity* *(continuous)*	0.84	0.05	0.74–0.95	0.007
*Belief that the vaccine offers high protection against COVID-19*				
Uncertain/Disagree	1 ª			
Agree	0.08	0.08	0.01–0.57	0.012
*Gender*				
Male	1 ª			
Female	0.05	0.06	0.004–0.61	0.019
*Need for further information on COVID-19 prevention or COVID-19 vaccine*				
No	1 ª			
Yes	11.55	12.09	1.48–89.84	0.019
*Self-rated health status*				
Low (1–7)	1 ª			
High (8–10)	0.13	.012	0.02–0.80	0.028
*Knowledge of all SARS-CoV-2 routes of transmission and main preventive measures*				
No	1 ª			
Yes	0.18	0.14	0.03–0.089	0.036
*Age in years (continuous)*	1.10	0.05	1.01–1.22	0.040
*Belief that the COVID-19 vaccine also protects family from* *contagion*				
Uncertain/Disagree	1 ª			
Agree	0.17	0.17	0.02–1.20	0.077
*Knowing main COVID-19 symptoms*				
Others	1 ª			
Fever/Cough/Tiredness	3.05	2.50	0.61–15.21	0.173
*Belief that use of masks in the workplace reduces risk of SARS-CoV-2 transmission*				
Uncertain/Disagree	1 ª			
Agree	3.18	3.94	0.28–36.09	0.349
*Having received information about COVID-19 prevention in the workplace from companies*				
No	1 ª			
Yes	0.53	0.036	0.013–2.07	0.364

ª Reference category; * Odds ratio; ** standard error; ◦ confidence interval.

## Data Availability

The data presented in this study are available on request from the corresponding author.

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
