# Peer review of "Preventive Measures for SARS-CoV-2 in the Workplace and Vaccine Acceptance: Assessment of Knowledge, Attitudes and Behaviors of Workers in Southern Italy"

_vaccines, 2022, doi:10.3390/vaccines10111872_

Round 1

Reviewer 1 Report

The paper is generally extremely well written, and I have read it with interest. The job is thoroughly and accurately detailed. The topic is properly introduced, the methods are appropriately chosen, and the results are sufficiently presented and discussed.

The biggest concern is the sample size (workers) in the study. Could you elaborate a bit about representativity? The discussion could confront the study with more references, as well.

In summary, the paper is organized well and easy to follow. 

Author Response

Reviewer 1

  1. The biggest concern is the sample size (workers) in the study. Could you elaborate a bit about representativity?

As suggested, we have included in the Discussion section a paragraph regarding the representativeness of the sample and the generalizability of the results of the survey.

  1. The discussion could confront the study with more references, as well.

As suggested, we have included in the Discussion section other comparisons of our findings with those of similar studies previously conducted.

Reviewer 2 Report

A very interesting and informative manuscript; however, there are some writing/editing issues that the authors should consider and address.  The following are suggestions/comments regarding these issues.  Line 18, "routes of transmission along with main preventive ...".  Line 44, "Severe Acute Respiratory Syndrome Coronavirus 2 (SARS-CoV-2) in the ...".  Lines 45 & 46, "...the adoption of Corona Virus Disease 19 (COVID-19) prevention and ...".  Line 52, "...in the workplaces" to be applied ...".  Line 59, "workers' health; therefore, in particular ...".  Line 68, "...professional categories [14-16].  Whereas, to the best ...".  Line 80, "...geographic area of the Campania region, ...".  Line 83, "...considered eligible.  Thereafter, ten companies ...".  Line 91, "...a preliminary meeting.  The objectives and the ...".  Lines 99 & 100, "...on other populations [17-21] and ...".  Line 101, "...of four sections: first, there were open ...".  Line 107, "...in the workplace, and comorbidities."  Line 113, "...preventive measures, flu and COViD-19 ...  Line 118, "...vaccination, with a multiple choice response format ...".  Line 125, "standard deviations.  The bivariate analysis ...".  Line 153, "... 50 to 250 employees), large ones ...".  Line 154, "employees) and belonged to the agri-food ...".  Lines 194 & 195, "...regression analysis, factors significantly influencing a higher knowledge about ...".  Line 208, "workplace.  73.3% believed that ...".  Line 209, "and 74.1% felt that a vaccine offers high protection ...".  Line 224, "...regression analysis, factors associated with ...".  Line 230, "...CI=1.14-3.76), and believing that ...".  Line 232, "...had received a flu vaccination in the ...".  Line 233, "...season and 47.5% were against COVID-19 at the ...".  Lines 237 & 238, "...is safe (5.7%), and perceiving a sense of ...".  Line 243, "...vaccine is effective (2.4%), and not being entitled ...".  Line 250, "...CI=0.74-0.95), not believing that physical distancing ...".  Line 251, "...workplace reduces the risk of ...".  Lines 251 & 252, "...CI=0.002-0.19) not believing that the vaccine ...".  Line 253, "CI=0.01-0.57) and needing further information on COVID-19 prevention or the vaccine ...".  Line 254, "...CI=1.48-89.84), and were not willing to ...".  Line 256, "...for COVID-19 and expressed their ...".  Line 257, "...reported reasons that they did not believe ...".  Lines 267 & 268, "...from media, 43.2% from physicians and 1.7% from companies."  Line 279, "...soon as the vaccine was available for their ...".  Lines 284 & 285, "...September 2020).  Working people were found to be more ...".  Line 289, "...professionals and teachers, and 44.8% among ...".  Line 291, "...to vaccination in the Italian territory, and 2) an increase in ...".  Lines 296 & 297, "...of the respondents recalled the main ones."  Line 302, "...Italian workers; 80.6% always used ...".  Line 303, "...sanitized their hands, and 35.8% avoided crowded ...".  Line 310, "...studies were conducted.  The present study being carried...".  Line 313, "...findings of Zhang et al. [14] highlighted vaccine ...".  Line 318, "...Zhang et al. in regards to the willingness of ...  Line 319, "...influenza vaccination status.  Different findings emerged ...".  Lines 320 & 321, "...according to Zhang, correlated with a higher intention to ...".  Line 323, "... received a COVID-19 vaccine of the factory ...".  Line 343, "This survey has some intrinsic limitations of ...".  Lines 347 & 348, "...and behaviors.  To minimize this, the questions were formulated in a way that did not influence in any way the respondents, especially in regards to reasons behind ...".  Line 349, "Recall bias and telescoping bias in particular could ...".  Lines 350 & 351, "...unpredictable manners, the results.  Also, it was not properly addressed for questionnaire limitations, involving the relation between ...".  Line 352, "...spreading of SARS-CoV-2 in the workplace."  Line 354 & 355, "...COVID-19 prevention and general knowledge."  Line 363, "...safety during a pandemic and to appropriately prepare ...".

Author Response

Reviewer 2

The following are suggestions/comments regarding these issues.  Line 18, "routes of transmission along with main preventive ...".  Line 44, "Severe Acute Respiratory Syndrome Coronavirus 2 (SARS-CoV-2) in the ...".  Lines 45 & 46, "...the adoption of Corona Virus Disease 19 (COVID-19) prevention and ...".  Line 52, "...in the workplaces" to be applied ...".  Line 59, "workers' health; therefore, in particular ...".  Line 68, "...professional categories [14-16].  Whereas, to the best ...".  Line 80, "...geographic area of the Campania region, ...".  Line 83, "...considered eligible.  Thereafter, ten companies ...".  Line 91, "...a preliminary meeting.  The objectives and the ...".  Lines 99 & 100, "...on other populations [17-21] and ...".  Line 101, "...of four sections: first, there were open ...".  Line 107, "...in the workplace, and comorbidities."  Line 113, "...preventive measures, flu and COViD-19 ...  Line 118, "...vaccination, with a multiple choice response format ...".  Line 125, "standard deviations.  The bivariate analysis ...".  Line 153, "... 50 to 250 employees), large ones ...".  Line 154, "employees) and belonged to the agri-food ...".  Lines 194 & 195, "...regression analysis, factors significantly influencing a higher knowledge about ...".  Line 208, "workplace.  73.3% believed that ...".  Line 209, "and 74.1% felt that a vaccine offers high protection ...".  Line 224, "...regression analysis, factors associated with ...".  Line 230, "...CI=1.14-3.76), and believing that ...".  Line 232, "...had received a flu vaccination in the ...".  Line 233, "...season and 47.5% were against COVID-19 at the ...".  Lines 237 & 238, "...is safe (5.7%), and perceiving a sense of ...".  Line 243, "...vaccine is effective (2.4%), and not being entitled ...".  Line 250, "...CI=0.74-0.95), not believing that physical distancing ...".  Line 251, "...workplace reduces the risk of ...".  Lines 251 & 252, "...CI=0.002-0.19) not believing that the vaccine ...".  Line 253, "CI=0.01-0.57) and needing further information on COVID-19 prevention or the vaccine ...".  Line 254, "...CI=1.48-89.84), and were not willing to ...".  Line 256, "...for COVID-19 and expressed their ...".  Line 257, "...reported reasons that they did not believe ...".  Lines 267 & 268, "...from media, 43.2% from physicians and 1.7% from companies."  Line 279, "...soon as the vaccine was available for their ...".  Lines 284 & 285, "...September 2020).  Working people were found to be more ...".  Line 289, "...professionals and teachers, and 44.8% among ...".  Line 291, "...to vaccination in the Italian territory, and 2) an increase in ...".  Lines 296 & 297, "...of the respondents recalled the main ones."  Line 302, "...Italian workers; 80.6% always used ...".  Line 303, "...sanitized their hands, and 35.8% avoided crowded ...".  Line 310, "...studies were conducted.  The present study being carried...".  Line 313, "...findings of Zhang et al. [14] highlighted vaccine ...".  Line 318, "...Zhang et al. in regards to the willingness of ...  Line 319, "...influenza vaccination status.  Different findings emerged ...".  Lines 320 & 321, "...according to Zhang, correlated with a higher intention to ...".  Line 323, "... received a COVID-19 vaccine of the factory ...".  Line 343, "This survey has some intrinsic limitations of ...".  Lines 347 & 348, "...and behaviors.  To minimize this, the questions were formulated in a way that did not influence in any way the respondents, especially in regards to reasons behind ...".  Line 349, "Recall bias and telescoping bias in particular could ...".  Lines 350 & 351, "...unpredictable manners, the results.  Also, it was not properly addressed for questionnaire limitations, involving the relation between ...".  Line 352, "...spreading of SARS-CoV-2 in the workplace."  Line 354 & 355, "...COVID-19 prevention and general knowledge."  Line 363, "...safety during a pandemic and to appropriately prepare ...".

As suggested, we have made the required changes.